# The Prognostic Effect of *KRAS* Mutations in Non-Small Cell Lung Carcinoma Revisited: A Norwegian Multicentre Study

**DOI:** 10.3390/cancers13174294

**Published:** 2021-08-26

**Authors:** Sissel Gyrid Freim Wahl, Hong Yan Dai, Elisabeth Fritzke Emdal, Thomas Berg, Tarje Onsøien Halvorsen, Anine Larsen Ottestad, Marius Lund-Iversen, Odd Terje Brustugun, Dagny Førde, Erna-Elise Paulsen, Tom Donnem, Sigve Andersen, Bjørn Henning Grønberg, Elin Richardsen

**Affiliations:** 1Department of Clinical and Molecular Medicine, NTNU, Norwegian University of Technology and Science, N-7491 Trondheim, Norway; hong.yan.dai@stolav.no (H.Y.D.); tarje.onsoien.halvorsen@stolav.no (T.O.H.); anine.l.ottestad@ntnu.no (A.L.O.); bjorn.h.gronberg@ntnu.no (B.H.G.); 2Department of Pathology, St. Olav’s Hospital, Trondheim University Hospital, N-7006 Trondheim, Norway; elisabeth.fritzke.emdal@stolav.no; 3Department of Clinical Pathology, University Hospital of North Norway, N-9038 Tromsø, Norway; thomas.berg@unn.no (T.B.); elin.richardsen@unn.no (E.R.); 4Department of Medical Biology, UiT, The Arctic University of Norway, N-9011 Tromsø, Norway; 5Department of Oncology, St. Olav’s Hospital, Trondheim University Hospital, N-7030 Trondheim, Norway; 6Department of Pathology, Oslo University Hospital, The Norwegian Radium Hospital, N-0310 Oslo, Norway; mlundive@ous-hf.no; 7Department of Cancer Genetics, Institute for Cancer Research, Oslo University Hospital, The Norwegian Radium Hospital, N-0450 Oslo, Norway; Odd.Terje.Brustugun@rr-research.no; 8Section of Oncology, Drammen Hospital, Vestre Viken Hospital Trust, N-3004 Drammen, Norway; 9Department of Clinical Medicine, UiT, The Arctic University of Norway, N-9037 Tromsø, Norway; dagnynymo@gmail.com (D.F.); tom.donnem@uit.no (T.D.); sigve.andersen@uit.no (S.A.); 10Department of Pulmonary Medicine, University Hospital of North Norway, N-9028 Tromsø, Norway; erna-elise.paulsen@unn.no; 11Department of Oncology, University Hospital of North Norway, N-9038 Tromsø, Norway

**Keywords:** non-small cell lung cancer, cohort study, survival, signalling pathway, *KRAS*, *KRAS* G12C

## Abstract

**Simple Summary:**

In this multicentre study of 1117 patients with stage I–IV non-squamous non-small cell lung carcinoma (NSCLC), we investigated associations between *KRAS* and clinical characteristics and survival. We investigated survival among the following groups of patients: those with no *KRAS* mutations (wild type) versus those with mutated tumours, those with *KRAS* wild type versus *KRAS* G12C versus *KRAS* non-G12C mutated tumours and among patients with different *KRAS* mutation subtypes. We also grouped *KRAS* mutated patients according to mutation preference for the Raf, PI3K/Akt and RalGDS/Ral intracellular signalling pathways and investigated whether there were differences in survival according to their preference for these pathways. We found that the proportion of *KRAS* mutated patients and frequency of *KRAS* mutation subtypes in our study is consistent with other studies of non-Asian patients with NSCLC. In multivariable analyses, we found no significant differences in the time to disease progression or overall survival within any of the analysed groups.

**Abstract:**

Background: due to emerging therapeutics targeting *KRAS* G12C and previous reports with conflicting results regarding the prognostic impact of *KRAS* and *KRAS* G12C in non-small cell lung cancer (NSCLC), we aimed to investigate the frequency of *KRAS* mutations and their associations with clinical characteristics and outcome. Since mutation subtypes have different preferences for downstream pathways, we also aimed to investigate whether there were differences in outcome according to mutation preference for the Raf, PI3K/Akt, or RalGDS/Ral pathways. Methods: retrospectively, clinicopathological data from 1233 stage I–IV non-squamous NSCLC patients with known *KRAS* status were reviewed. *KRAS*’ associations with clinical characteristics were analysed. Progression free survival (PFS) and overall survival (OS) were assessed for the following groups: *KRAS* wild type (wt) versus mutated, *KRAS* wt versus *KRAS* G12C versus *KRAS* non-G12C, among *KRAS* mutation subtypes and among mutation subtypes grouped according to preference for downstream pathways. Results: a total of 1117 patients were included; 38% had *KRAS* mutated tumours, 17% had G12C. Among *KRAS* mutated, G12C was the most frequent mutation in former/current smokers (45%) and G12D in never smokers (46%). There were no significant differences in survival according to *KRAS* status, G12C status, among *KRAS* mutation subtypes or mutation preference for downstream pathways. Conclusion: *KRAS* status or *KRAS* mutation subtype did not have any significant influence on PFS or OS.

## 1. Introduction

Mutations of the *v-Ki-ras2 Kirsten rat sarcoma viral oncogene homology gene* (*KRAS*) gene are the most common oncogenic drivers of non-squamous non-small cell lung carcinoma (NSCLC) and occur in approximately 25–38% of non-Asian and 8–10% of Asian lung adenocarcinoma patients [1,2,3]. Although associated with smoking, *KRAS* mutations also occur in approximately 5–15% of never-smoking patients [1,4]. *KRAS*’ role as a prognostic factor has been investigated in numerous studies, but with conflicting results [5,6,7,8,9,10,11]. This may be attributed to heterogeneity of the study populations regarding sample size, disease stage, ethnicity, histological subtypes, study end points, and therapeutic history. Importantly, evaluation of the prognostic value is further complicated by the diverse and complex biological effects of mutated *KRAS* in signal transduction. Co-occurring genetic alterations in other genes have also been shown to have an impact on survival, exemplified by co-mutations in *STK11* or *KEAP1*, which are associated with inferior survival compared to *KRAS* mutation only [2,12,13].

The *KRAS* gene encodes a small, cell-membrane bound guanosine triphosphate (GTP)ase, which is central in signal transduction through receptor tyrosine kinases via the Raf/Mek/Erk, PI3K/Akt, RalGDS-RalA/B and other signalling pathways. The Ras protein switches between an inactive guanosine diphosphate (GDP) bound and active GTP bound state [14]. The active state is promoted by a Ras guanine exchange factor, which enhances dissociation of GDP from Ras and Ras binding to GTP. GTPase activating protein (GAP) and the Ras protein’s intrinsic GTPase activity facilitate hydrolysis of GTP, returning Ras to its inactive GDP-bound state.

*KRAS* hot spot mutations are clustered on codon 12 and 13 in exon 2 and codon 61 in exon 3 [15]. The most common mutation subtypes in *KRAS* mutated lung adenocarcinoma are the codon 12 transversion mutations (substitution of a purine with a pyrimidine nucleotide, or opposite) G12C (39%) and G12V (18–21%), followed by the transition mutations (substitution of a purine by a purine, or a pyrimidine with a pyrimidine) G12D (14–18%) and G12A (10–11%) [1,2,16]. While *KRAS* transversion mutations are associated with a history of smoking, *KRAS* transition mutations are more common in never-smokers [1]. The oncogenic Ras proteins interfere with the GDP/GTP exchange and GTP hydrolysis, leaving the mutated Ras protein in a constitutively active GTP-bound state [17] with subsequent continuous activation of downstream pathways. The Ras oncoproteins may have some differences in affinity for downstream effector proteins. G12A, G12C, G13D, Q61L, and Q61H have been shown to have higher preference for Raf interaction [18]. G12C also has a high preference for RalGDS-RalA/B while G12D has been associated with preference for interaction with PI3K/Akt [19,20].

With the prospect of targeted treatment of patients with *KRAS* G12C mutated NSCLC, we retrospectively aimed to explore the frequency of *KRAS* mutations, clinical characteristics and the prognostic effects of *KRAS* in a cohort of patients diagnosed with non-squamous NSCLC disease stage I–IV in three university hospitals in Norway. We aimed to investigate potential differences in progression free survival (PFS) and overall survival (OS) in patients with *KRAS* wild type (wt) compared to those with *KRAS* mutated (mut) tumours, in patients with *KRAS* G12C compared to patients with *KRAS* wt and *KRAS* non-G12C mutations, and among *KRAS* mutation subtypes. Due to heterogeneity of biological effects of mutated Ras proteins, we also investigated whether survival was associated with *KRAS* mutation preference for interaction with either the PI3K/Akt, Raf- or RalGDS/Ral pathways.

## 2. Material and Methods

### 2.1. Ethics

This study was approved by the Regional Committees for Medical and Health Research Ethics (REC) in Eastern, Central, and Northern Norway (identification number 82144). The study also included patients enrolled in the regional research biobanks of Eastern, Central and Northern Norway. These research biobanks are approved by REC in Eastern, Central and Northern Norway, the Ministry of Health and Care Services and the Norwegian Data Protection Authority. Patients enrolled in the biobanks are over 18 years old and have given written informed consent.

### 2.2. Patient Inclusion and Tumour Specimens

Patients diagnosed with non-squamous non-small cell lung carcinoma (NSCLC) stage I–IV at St. Olav’s Hospital (STO, *n* = 676), the University Hospital of North Norway (UNN, *n* = 293) and Oslo University Hospital (OUH, *n* = 264) between 2003 and 2020 were evaluated for inclusion. Of these were 594 patients included in the regional biobanks (STO *n* = 266, UNN *n* = 64, OUH *n* = 264). Patients fulfilling all of the following criteria were included: (a) non-squamous (non-neuroendocrine), histology or immunophenotype; (b) known *KRAS* mutation status; (c) known mutational, rearrangement status of the *Epidermal Growth Factor Receptor (EGFR)*/*Anaplastic Lymphoma Receptor Tyrosine Kinase (ALK)*/*ROS Proto-Oncogene 1, Receptor Tyrosine Kinase (ROS1)* genes, or if status was unknown, treatment naïve to tyrosine kinase inhibitors (TKI); and (d) no non-pulmonary synchronous malignancy. Patients treated with curative intent for other malignancies and who were recurrence free ≥ 5 years before the lung cancer were also evaluated for the study. Patients with ≥2 pulmonary nodules at the time of diagnosis were excluded, unless the tumours were in the same lobe and of the same histology and same *EGFR*, *KRAS*, *ALK*, or *ROS1* status. The following information was retrieved from the hospital medical records and pathology reports: age, sex, smoking history (current smoker, former smoker > 1 year prior to diagnosis or never smoker), Eastern Cooperative Oncology Group performance status (ECOG PS), pathological disease stage if surgical treatment, clinical disease stage if no surgery, extent of disease and metastatic sites at the time of diagnosis, results of molecular analyses, first line tumour treatment, history of second and later treatment lines, history of treatment with TKI or immune checkpoint inhibitors (ICI), date of first relapse, and date of death. All tumour specimens were reviewed and classified according to the fourth edition of the WHO Classification of Lung Tumours [21] by experienced lung pathologists (authors S.G.F.W., E.R., M.L.I.) in the respective pathology departments of the three hospitals. Clinical or pathologic restaging was performed according to the eighth edition of The New American Joint Committee on Cancer/International Union Against Cancer TNM stage classification for lung cancer [22].

Tumour specimens were analysed for *KRAS* mutations either by mutation specific real time PCR targeting codon 12 and 13 of exon 2 and codon 61 of exon 3 (OUH only) according to protocols implemented for routine diagnostics in the respective pathology departments at STO and OUH, or by next generation sequencing (NGS). NGS was performed using Illumina TruSight^®^ Tumour 15 and TruSight^®^ Tumour 26 (Illumina^®^, San Diego, CA, USA; used at UNN and STO, respectively) or QIAseq^TM^ Comprehensive Targeted DNA Panel (Qiagen, Hilden, Germany; used at STO).

### 2.3. Statistics

The chi-square test for independence was used for comparison of categorical variables. PFS was defined as the time from the first diagnostic tissue specimen (biopsy or cytology) until objective progression or death by any cause. OS was defined as the time from the first diagnostic tissue specimen until death by any cause. Median follow-up time for PFS and OS was estimated using the reversed Kaplan–Meier method. Survival was estimated using the Kaplan–Meier method and compared using the log-rank test. The Cox proportional hazards model was used for univariable and multivariable analyses. The significance level was defined as a two-sided *p* < 0.05. All analyses were performed using the IBM SPSS Statistics for Windows version 27.0. (Armonk, NY, USA: IBM Corp.).

## 3. Results

### 3.1. Patient Characteristics

Of the 1233 patients evaluated for inclusion, 1117 were eligible for this study (Figure 1). Of these, 622 (55.7%) were diagnosed with NSCLC at STO, 264 (23.6%) at OUH and 231 (20.7%) at UNN.

Patients’ characteristics according to *KRAS* mutation status are presented in Table 1. Median age was 69 (range 32–90) years, 592 (53%) were women, 950 (89%) were former or current smokers, 1063 (95%) were diagnosed with adenocarcinoma; ECOG PS was 0–1 in 1037 (93%) patients. The distribution of patients according to disease stage I-IV was 359 (32%), 148 (13%), 230 (21%), and 380 (34%), respectively.

Of the 1117 patients, 46 (4%) had no treatment due to comorbidities, 671 (60%) had potentially curative treatment and of these had 572 (85%) complete surgical resection. Of the 400 (36%) patients with advanced disease treated with palliative intention, the dominant first line treatments were platinum-doublet chemotherapy in 137 (34%) and radiochemotherapy in 91 (23%). Detailed overviews of curative and palliative treatments are presented in Appendix A. Of the 1117 patients included, 420 (38%) had *KRAS* mut tumours, 142 (13%) had *EGFR* mut tumours, 12 (1%) had *ALK* rearranged tumours, and 3 (0.3%) patients had *ROS1* rearranged tumours. *EGFR*, *ALK*, and *ROS1* analyses were not performed in 27, 94, and 562 of the patients, respectively, since these analyses were not routinely performed in Norway at the time of diagnosis. None of the patients with unknown *EGFR*/*ALK*/*ROS1* status was treated with TKIs.

### 3.2. KRAS Mutation Status and Correlations with Clinical Characteristics

An overview of associations between patient characteristics and *KRAS* status is presented in Appendix A. Among patients with *KRAS* mutated tumours, 407 (97%) were current or former smokers and 13 (3%) were never smokers (*p* < 0.001). The proportion of women with *KRAS* mutated tumours was higher compared to men (57% versus 43%, respectively, *p* = 0.042). There were no associations between the presence of *KRAS* mutation and age, ECOG PS, disease stage, treatment history of surgery and number of metastatic sites at the time of diagnosis. At the time of diagnosis, the proportion of patients with pleural metastases, as either the only metastatic site or concurrent with other metastatic sites, was higher for patients with *KRAS* wt tumours (13.1%) than patients with *KRAS* mut tumours (6.5%, *p* < 0.001). No associations between *KRAS* status and metastases in other sites (adrenal gland, liver, skeleton and brain) were found. We found no associations between *KRAS* G12C, G12V or G12D and clinical characteristic.

In the whole cohort of 1117 patients, 192 (17%) had *KRAS* G12C, 81 (7%) G12V, 70 (6%) G12D, and 30 (3%) G12A. The frequencies of *KRAS* mutation subtypes are presented in Figure 2.

Within the group of patients with *KRAS* mut tumours, G12C was the most frequent mutation in former/current smokers (45%), while G12D was more frequent in *KRAS* mut never smokers (46%; *p* = 0.016; Appendix A). Among patients with the three most common *KRAS* mutation subtypes, *KRAS* G12C, G12V, and G12D, there were no significant differences in distribution of mutation subtype according to sex, age, disease stages, surgical history, and the number of metastatic sites or metastatic site at time of diagnosis (Appendix A).

### 3.3. Mutation Status and Survival

Median follow-up for PFS was 52.7 (95%CI 44.3–61.2) months and for OS 52.7 (95%CI 45.7–59.6) months; 419 patients were progression-free, and 547 patients were alive at the time of data completion (April 2020). In the whole cohort, estimated median PFS was 17.2 (95%CI 13.6–20.7) months and estimated median OS 38.1 (95%CI 30.1–46.0) months.

#### 3.3.1. Whole Cohort (Stage I–IV)

In the univariable analyses, neither *KRAS* status, G12C status (G12C versus *KRAS* wt versus *KRAS* non-G12C), nor *KRAS* mutation subtypes grouped according to signalling pathway preference, had effect on PFS or OS. Among *KRAS* mutation subtypes, G12C was associated with an effect on PFS (HR 0.62; 95%CI 0.38–1.00, *p* = 0.050) and OS (HR 0.59; 95%CI 0.035–0.99, *p* = 0.044) compared to G12A, but not to G12V or G12D. An overview of univariable analyses of relations between all covariates and survival is presented in Table 2.

There were no differences in estimated median PFS or OS between patients with *KRAS* wt/*KRAS* mut and between patients with *KRAS* wt, *KRAS* G12C, or *KRAS* non-G12C, neither in the log-rank tests (Figure 3) or multivariable analyses adjusting for age, sex, smoking history, ECOG PS, disease stage and treatment type (Table 3).

Due to a trend towards better PFS for patients with G12C compared to patients with non-G12C *KRAS* mutations on pairwise log-rank test (*p* = 0.080), we further compared survival among patients with the four most frequent *KRAS* mutations (Figure 4). In these analyses, the estimated median PFS for G12C was 27.0 (95%CI 14.2–39.8) months compared to 16.3 (95%CI 10.9–21.7) months for G12V, 13.2 (95%CI 9.2–17.3) for G12D and 8.5 (95%CI 3.3–13.8) months for G12A (*p* = 0.218). The pairwise log-rank test showed significantly better PFS for patients with G12C compared to G12A (*p* = 0.042), but not G12V (*p* = 0.329) or G12D (*p* = 0.311). The estimated median OS was 57.6 (95%CI 28.6–86.5) months for G12C, 49.1 (95%CI 5.6–92.7) months for G12V, 34.8 (95%CI 0–76.4) months for G12D and 18.5 (95%CI 1.3–35.8) months for G12A. On pairwise comparison, patients with G12C also had better OS compared to patients with G12A (*p* = 0.048), but not G12V (*p* = 0.895) or G12D (*p* = 0.384). The differences in PFS or OS between patients with G12C and G12A, however, did not remain statistically significant in multivariable analyses adjusting for age, sex, smoking history, ECOG PS, and treatment type (Table 4).

We then investigated PFS and OS in patients with mutation preference for the Raf pathway (G12C, G12A, G13D, Q61L, Q61H) versus patients with G12D favouring PI3K/Akt, and for patients with G12D favouring PI3K/Akt versus G12C favouring the Ral A/B pathway. We found no differences in estimated median PFS or OS between any of these groups, neither in the log-rank tests (Figure 5) or multivariable analyses (Table 5).

#### 3.3.2. Curative Surgery

There were no differences in PFS or OS between patients with *KRAS* mut/*KRAS* wt or *KRAS* wt/*KRAS* G12C/ *KRAS* non-G12C (Appendix A). We then compared survival among patients with G12C, G12V, G12D and G12A (Appendix A). Patients with G12C had longer estimated median PFS than G12V (88.4 versus 29.7 months, respectively), but was not estimated for G12D and G12A since the estimated survival probability did not reach 50% for these groups. The difference in estimated median PFS was significant for G12C compared to G12V in pairwise log-rank test (*p* = 0.037) but did not remain significant in multivariable analyses adjusting for sex, age, smoking history, treatment and disease stage (*p* = 0.180, Appendix A). There were no significant differences in OS between patients with G12C, G12V, and G12D, but median OS was only estimated for G12V (Appendix A).

#### 3.3.3. Non-Curative Treatment

No differences in the estimated median PFS or OS were observed between patients with *KRAS* wt/*KRAS* mut (Appendix A). In the *KRAS* wt/*KRAS* G12C/*KRAS* non-G12C subgroups (Appendix A), *KRAS* wt patients had better OS compared to *KRAS* G12C (9.6 versus 6.7 months) in the pairwise log-rank test (*p* = 0.047), but this association did not remain significant in multivariable analysis adjusting for sex, age, smoking history, ECOG PS, history of chemotherapy, history of TKI, history of ICI, and disease stage (Appendix A). There were no differences in PFS or OS between patients with G12C, G12V, G12D, and G12A; neither in the overall or pairwise log-rank tests or in multivariable analyses (Appendix A).

## 4. Discussion

In this retrospective multicentre study of 1117 patients with non-squamous NSCLC, we describe associations between *KRAS* status and various clinicopathological characteristics and survival. The presence of *KRAS* mutation was significantly associated with a history of smoking, with G12C being the most frequent mutation in former and current smokers and G12D the most common mutation in never smokers. We also found a significantly higher proportion of women with *KRAS* mut tumours compared to men. The associations with survival were investigated according to *K**RAS* status (*KRAS* wt versus *KRAS* mut), *KRAS* G12 status (*KRAS* wt versus *KRAS* G12C versus *KRAS* non-G12C mutations) and *KRAS* mutation type (G12C, G12V, G12D, and G12A). We found no associations with survival for any of the compared groups in the multivariable analyses, in the analyses of the whole cohort, separately for resected patients with curative disease, or for patients with advanced disease. Furthermore, we found no associations with survival in subgroup analyses of *KRAS* mut patients grouped, according to mutation preference for interaction with the PI3K/Akt, Raf- or Ral pathways.

Our study is one of the largest studies on the prognostic effect of *KRAS* in non-squamous NSCLC in all disease stages. Since many studies on the prognostic value of *KRAS* in NSCLC have focused on patients with either local or advanced disease, we also performed isolated subgroup analyses of patients with resected curative disease and advanced disease, in addition to the analyses of the whole cohort. Furthermore, to our knowledge we present the first study of patients with NSCLC where specific *KRAS* mutations and their preference for signalling pathways have been taken into consideration in survival analyses.

Regarding survival in patients with *KRAS* wt and *KRAS* mut tumours, our results are consistent with other studies [5,6,7,8,9,23]. Moreover, we found no significant differences in PFS or OS between patients with *KRAS* wt, *KRAS* G12C, and *KRAS* non-G12C mutated tumours or among patients with G12C, G12V, G12D, and G12A mutated tumours, which agrees with other studies [2,13,16,24,25,26,27]. However, worse survival in patients with *KRAS* mutated tumours (as one group) compared to *KRAS* wt, as well as in patients with *KRAS* G12C compared to patients with *KRAS* non-G12C mutations, have also been reported [2,10,11,28].

The conflicting results on *KRAS* as a prognostic factor may be attributed to several factors, including differences of the study populations as mentioned in the introduction. Cross-study comparison is also challenging due to differences in follow-up, definitions of endpoints and variability in covariates adjusted for in multivariable analyses.

Investigations of the prognostic impact of *KRAS* mutations may further be complicated by the diverse biological effects of the mutated Ras proteins. In addition to different preferences for signalling pathways, studies on cell lines have also shown that different K-Ras oncoprotein subtypes also have phenotypical biochemical differences in terms of GTP affinity, the ability to speed up the GDP to GTP exchange and the ability to reduce the speed of intrinsic and GAP mediated hydrolysis [18,29]. Hence, grouping patients according to pathway preference for survival analyses is a simplified approach.

Evaluation of the prognostic value of *KRAS* is also complicated by co-occurring mutations in other genes. The presence of concurrent mutations in *STK11* and *KEAP1* have been reported at frequencies of 12–29% and 8–27%, respectively, and have been associated with worse recurrence free -or overall survival compared to *KRAS* mutation only [2,12,13]. Concurrent genomic alterations of *KEAP1* and *CDKN2A* are also associated with reduced T-cell inflammation and low levels of PD-L1 expression, predictive of reduced response to immune checkpoint inhibitors [12,30,31,32,33]. *KRAS* mutated tumours with co-occurring mutations in the *TP53* gene (reported frequency of 39–42%), on the other hand, are associated with active inflammation, high expression of PD-L1 and increased response to ICI [12,13,30,33,34]. The study by Scheffler et al. [33] also indicates that co-occurring mutations in specific genes may be associated with specific *KRAS* mutation subtypes.

Other less studied mechanisms, including mutant allele specific imbalance (MASI) and expression patterns, may also have effects on survival. Villaruz et al. showed that high levels of *KRAS* mutated alleles compared to *KRAS* wild type alleles was associated with significant worse PFS [24]. Nagy et al. combined gene expression data in patients with *KRAS* mut adenocarcinomas and generated a gene expression signature based on the five strongest genes expressed secondary to *KRAS* mutation [35]. Patients with high gene signature expressions had significantly shorter OS compared to the *KRAS* mutated patients with low expression.

Taken together, the traditional approaches to evaluate the prognostic value of mutated *KRAS* in mixed groups comprising patients with different mutation subtypes with different biological properties, may be too narrow. It is an increasing understanding that *KRAS* mutated NSCLCs are genetically heterogenous diseases. Hence, the complex biological diversity of *KRAS* mutated NSCLC should be taken into consideration when exploring associations with clinicopathological characteristics and outcome

There are some limitations to our study. These include the retrospective nature of the study. In our subgroup analyses of mutation preference for signalling pathways, we included G12C, G12A, G13D, and Q61L/H in the group “favouring Raf”, since these mutations were found to have high affinity for Raf in the study by Hunter et al. [18]. However, when considering the lower intrinsic hydrolysis rates for KRAS Q61L and G12A compared to G12C and G13D in this study, KRAS Q61L and G12A were predicted to be stronger activators of Raf. Due to the low number of patients with *KRAS* G12A and Q61L mutations in our study, we also included G12C and G12D in the group “favouring Raf”.

Another limitation is that we did not perform any additional molecular analyses to explore differences in expression of the main targets of the mutant Ras proteins. In a recent study of patients with *KRAS* G12C mutated colon cancer, it was shown that comprehensive analyses of gene expression profiles, co-occurring alterations of other genes and protein expression might shed light on the involvement of signalling pathways [36].

We also wanted to explore whether *KRAS* was of predictive value in patients with local disease treated with stereotactic body radiation therapy or conventional radiotherapy 60–66 Gy, but the number of these patients in our cohort was too small for analyses. Comprehensive next generation sequencing was only performed for a small subset of the patients. Hence, we did not have sufficient molecular data on co-occurring mutations in other genes, including *STK11*, *KEAP1, CDKN2A*, and *TP53*, for exploration of clinicopathological associations, associations with specific *KRAS* mutation subtypes, the prognostic value of concurrent genetic alterations or the predictive value with respect to ICI therapy. It would also be of interest to explore associations between *KRAS* mutation subtypes and tumour expression of programmed death ligand 1 (PD-L1), but we did not collect information on PD-L1 expression.

Although *KRAS* was the first oncogene to be associated with NSCLC [37], designing targeted therapies targeting the mutated K-Ras proteins has proven to be challenging due to the complex biology of the oncogenic Ras proteins and their high affinity for GTP. However, the development of G12C inhibitors which irreversibly bind to cysteine in the mutant G12C, locking the protein in an inactive GDP-bound state, have shown promising results in recent clinical phase I and II trials [38,39]. G12C inhibitors may improve the treatment options for a substantial proportion of patients with non-squamous NSCLC, with subsequent increased interest in the prognostic value of *KRAS* G12C. However, recent reports suggest a diversity of molecular alterations and mechanisms conferring adaption and resistance to G12C inhibitors [40,41]. Comprehensive molecular testing beyond *KRAS* mutation subtype may therefore be warranted before and during treatment with G12C inhibitors to identify possible alterations conferring resistance.

## 5. Conclusions

In this multicentre study of patients with non-squamous NSCLC, we found no differences in PFS or OS between patients with *KRAS* mutated and *KRAS* wild type NSCLC, between patients with *KRAS* wild type, G12C and *KRAS* non-G12 mutations, or among *KRAS* mutation subtypes. Furthermore, we found no differences in survival among patients grouped according to their mutation’s preference for either Raf, PI3K/Akt, or Ral pathways.

## Figures and Tables

**Figure 1 cancers-13-04294-f001:**
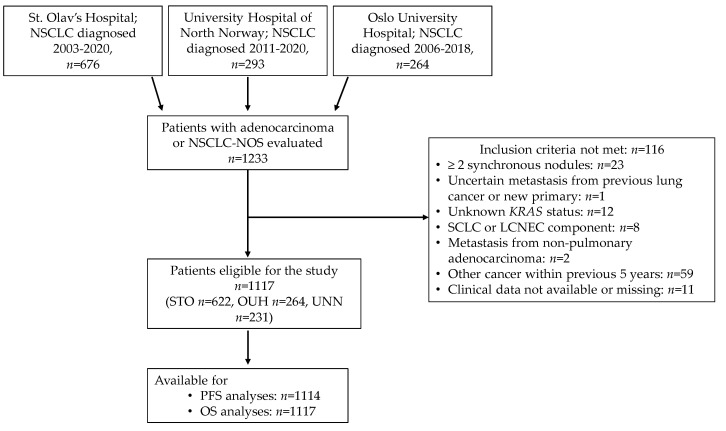
Outline of patient selection. Abbreviations: NSCLC, non-small cell lung carcinoma; NOS, not otherwise specified; STO, St. Olav’s Hospital; UNN, University Hospital of North Norway; OUH, Oslo University Hospital; SCLC, small cell lung carcinoma; LCNEC, large cell neuroendocrine carcinoma; PFS, progression free survival; OS, overall survival.

**Figure 2 cancers-13-04294-f002:**
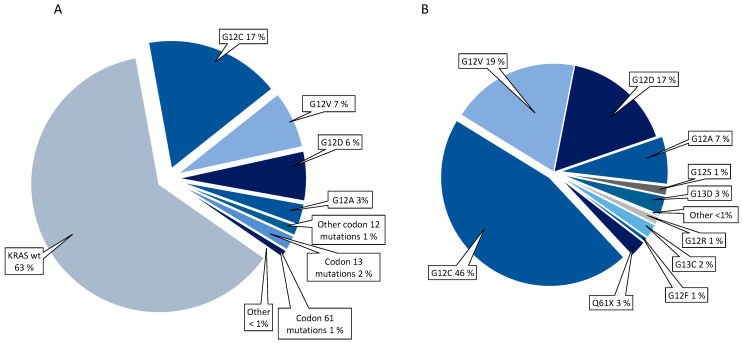
Frequencies of KRAS mutations in the whole cohort (**A**) and within the group of *KRAS* mutated patients (**B**).

**Figure 3 cancers-13-04294-f003:**
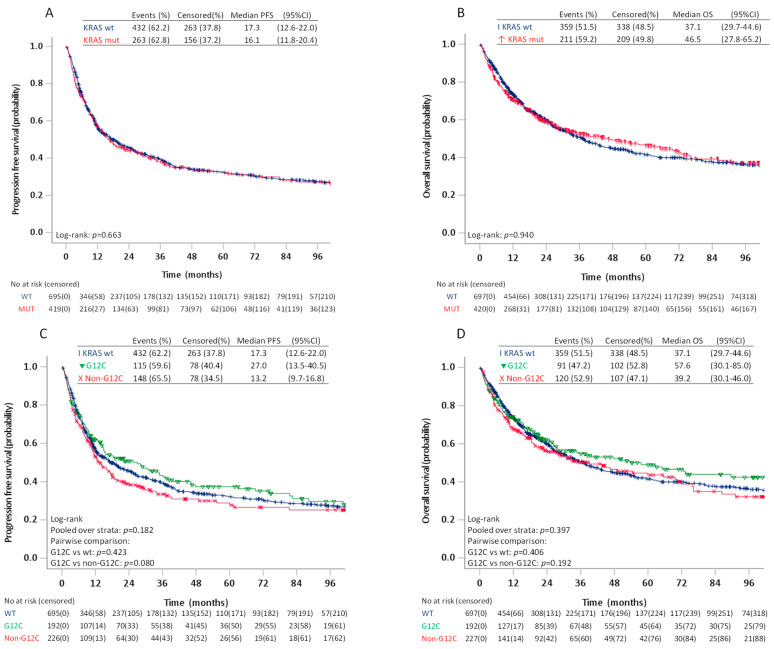
Progression free survival and overall survival in patients stage I–IV with KRAS wild type and KRAS mutated tumours (**A**,**B**) and patients with KRAS wild type, KRAS G12C, and KRAS non-G12C (**C**,**D**).

**Figure 4 cancers-13-04294-f004:**
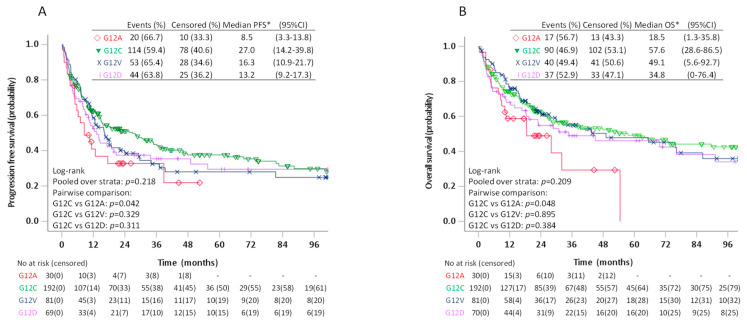
Progression free survival (**A**) and overall survival (**B**) in patients stage I–IV with KRAS G12C, G12V, G12D, and G12A mutated tumours.

**Figure 5 cancers-13-04294-f005:**
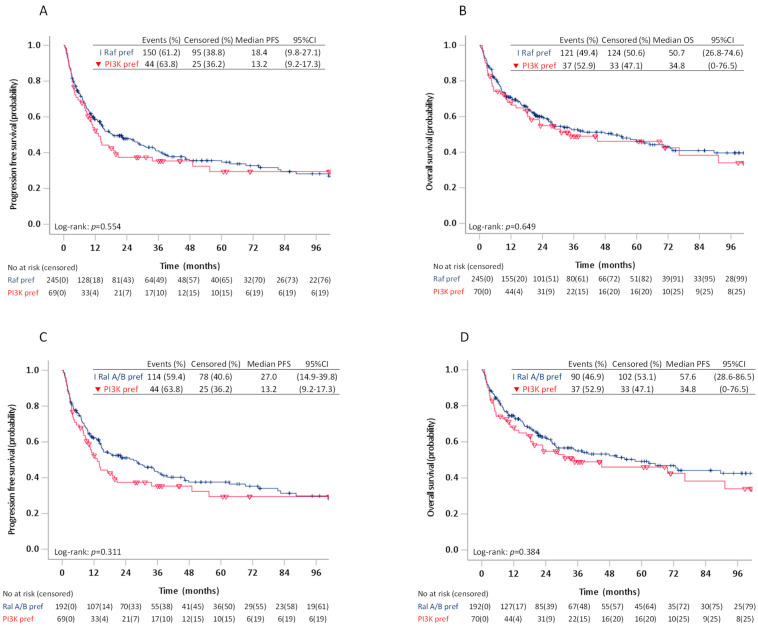
Progression free survival and overall survival in patients stage I–IV with KRAS mutants favouring the Raf pathway (G12C, G12A, G13D, Q61L/H) versus patients with KRAS G12D favouring PI3K/Akt (**A**,**B**) and patients with KRAS G12C favouring the RalA/B pathway versus patients with KRAS G12D favouring PI3K/Akt (**C**,**D**).

**Table 1 cancers-13-04294-t001:** Baseline patient characteristics according to KRAS status.

Characteristic	Total		KRAS wt *n* = 697		KRAS mut *n* = 420	
Age (years)						
<50	46	(4.1)	29	(4.2)	17	(4.0)
50–60	169	(15.1)	104	(14.9)	65	(15.5)
>60	902	(80.8)	564	(80.9)	338	(80.5)
Hospital						
STO	622	(55.7)	391	(56.1)	231	(55.0)
UNN	231	(20.7)	144	(20.7)	87	(20.7)
OUH	264	(23.6)	162	(23.2)	102	(24.3)
Sex						
Female	592	(53.0)	353	(50.6)	239	(56.9)
Male	525	(47.0)	344	(49.4)	181	(43.1)
Smoking history						
Never smoker	126	(11.3)	113	(16.2)	13	(3.1)
Former/current smoker	991	(88.7)	584	(83.8)	407	(96.9)
Ethnicity						
Scandinavian/European	1106	(99.0)	686	(98.4)	420	(100.0)
African	4	(0.4)	4	(0.6)	0	(0.0)
Asian	7	(0.6)	7	(1.0)	0	(0.0)
ECOG PS						
0–1	1037	(92.8)	650	(93.3)	387	(92.1)
2	59	(5.3)	35	(5.0)	24	(5.7)
3–4	21	(1.9)	12	(1.7)	9	(2.1)
Histology						
Adenocarcinoma	1063	(95.2)	650	(93.3)	413	(98.3)
Adenosquamous carcinoma	6	(0.5)	6	(0.9)	0	(0.0)
Large cell carcinoma	3	(0.3)	3	(0.4)	0	(0.0)
NSCLC-NOS	36	(3.2)	31	(4.4)	5	(1.2)
MIA	2	(0.2)	2	(0.3)	0	(0.0)
Other non-squamous NSCLC	7	(0.6)	5	(0.7)	2	(0.5)
EGFR status						
No	948	(84.9)	528	(75.8)	420	(100.0)
Yes	142	(12.7)	142	(20.4)	0	(0.0)
Not assessed	27	(2.4)	27	(3.9)	0	(0.0)
ALK rearrangement						
No	1011	(90.5)	620	(89.0)	391	(93.1)
Yes	12	(1.1)	12	(1.7)	0	(0.0)
Not assessed	94	(8.4)	65	(9.3)	29	(6.9)
ROS1 rearrangement						
No	552	(49.4)	342	(49.1)	210	(50.0)
Yes	3	(.3)	3	(0.4)	0	(0.0)
Not assessed	562	(50.3)	352	(50.5)	210	(50.0)
Disease stage						
IA	227	(20.3)	141	(20.2)	86	(20.5)
IB	132	(11.8)	77	(11.0)	55	(13.1)
IIA	40	(3.6)	19	(2.7)	21	(5.0)
IIB	108	(9.7)	75	(10.8)	33	(7.9)
IIIA	138	(12.4)	83	(11.9)	55	(13.1)
IIIB	67	(6.0)	42	(6.0)	25	(6.0)
IIIC	25	(2.2)	16	(2.3)	9	(2.1)
IVA	206	(18.4)	140	(20.1)	66	(15.7)
IVB	174	(15.6)	104	(14.9)	70	(16.7)
Treatment intention						
Curative	671	(60.1)	414	(59.4)	257	(61.2)
Palliative	400	(35.8)	251	(36.0)	149	(35.5)
No treatment	46	(4.1)	32	(4.6)	14	(3.3)

Abbreviations: wt, wild type; mut, mutated; STO, St. Olav’s Hospital; UNN, University Hospital of North Norway; OUH, Oslo University Hospital; ECOG PS, Eastern Cooperative Oncology Group performance status; NSCLC-NOS, non-small cell lung carcinoma, not otherwise specified; MIA, minimal invasive adenocarcinoma.

**Table 2 cancers-13-04294-t002:** Univariable analyses (Cox proportional hazards model) of patient and clinical characteristics and their associations with PFS and OS in the whole cohort.

Variable	PFS				OS			
	*n*	HR	95% CI	*p*	*n*	HR	95% CI	*p*
Age (years)								
<50	46	1 (ref)			46	1 (ref)		
50–60	166	1.07	0.71–1.60)	0.762	169	1.49	(0.90–2.45)	0.118
>60	902	1.04	(0.72–1.51)	0.818	902	1.68	(1.06–2.67)	0.027
Sex								
Women	590	1 (ref)			592	1 (ref)		
Men	524	1.10	(0.94–1.27)	0.233	525	1.13	(0.96–1.34)	0.137
Smoking history								
Never	126	1 (ref)			126	1 (ref)		
Former/current	988	1.48	(1.15–1.91)	0.002	991	1.59	(1.19–2.12)	0.002
ECOG PS								
0–1	1034	1 (ref)			1037	1 (ref)		
2	59	4.30	(3.24–5.71)	<0.001	59	6.41	(4.80–8.56)	<0.001
3–4	21	4.55	(2.76–7.48)	<0.001	21	6.91	(4.18–11.41)	<0.001
Stage								
I	359	1 (ref)			359	1 (ref)		
II	148	2.15	(1.61–2.86)	<0.001	148	2.11	(1.51–2.95)	<0.001
III	230	4.37	(3.43–5.57)	<0.001	230	3.89	(2.93–5.15)	<0.001
IV	377	12.74	(10.01–16.09)	<0.001	380	12.46	(9.61–16.14)	<0.001
Surgery								
No	534	1 (ref)			534	1 (ref)		
Yes	580	0.15	(0.12–0.17)	<0.001	583	0.13	(0.11–0.16)	<0.001
Curative RT +/− CT first line								
No	1023	1 (ref)			1023	1 (ref)		
Yes	91	1.28	(0.99–1.64)	0.060	91	1.05	(0.78–1.43)	0.740
Palliative CT and/or RT first line								
No	843	1 (ref)			846	1 (ref)		
Yes	271	5.75	(4.85–6.81)	<0.001	271	5.54	((4.64–6.63))	<0.001
History of TKI (any line)								
No	1025	1 (ref)			1028	1 (ref)		
Yes	89	1.56	(1.22–1.99)	<0.001	89	1.32	(0.86–2.03)	0.198
History of ICI (any line)								
No	939	1 (ref)			941	1 (ref)		
Yes	175	2.19	(1.81–2.64)	<0.001	176	1.39	((1.11–1.74))	0.006
KRAS status								
Wild type	695	(1 (ref)			697	1 (ref)		
Mutated	419	1.04	(0.89–1.21)	0.664	420	1.01	(0.85–1.19)	0.940
KRAS G12C status								
Wild type	695	1 (ref)			697	1 (ref)		
G12C	192	0.91	(0.74–1.12)	0.379	193	0.91	(0.72–1.14)	0.414
KRAS non-G12C	227	1.15	(0.96–1.39)	0.132	227	1.10	(0.89–1.35)	0.383
Raf vs. PI3K preference								
Raf	245	1 (ref)			245	1 (ref)		
PI3K	69	1.11	(0.79–1.55)	0.554	70	1.09	(0.75–1.58)	0.649
Ral A/B vs. PI3K preference								
RalA/B	192	1 (ref)			192	1 (ref)		
PI3K	69	1.20	(0.85–1.70)	0.312	70	1.19	(0.81–1.74)	0.385
Type KRAS mutation								
G12A	30	1 (ref)			30	1 (ref)		
G12C	192	0.62	(0.38–1.00)	0.050	192	0.59	(0.35–0.99)	0.044
G12V	81	0.72	(0.43–1.21)	0.220	81	0.60	(0.34–1.06)	0.080
G12D	69	0.74	(0.44–1.26)	0.274	70	0.69	(0.39–1.24)	0.216

Abbreviations: PFS, progression free survival; OS, overall survival; HR, hazard ratio; ref reference; ECOG PS, Eastern Cooperative Oncology Group performance status; CT, chemotherapy; RT, radiotherapy; TKI, tyrosine kinase inhibitor; ICI, immune checkpoint inhibitor.

**Table 3 cancers-13-04294-t003:** Multivariable analyses of progression free survival and overall survival in patients st. I–IV with KRAS wild type, KRAS mutated, KRAS G12C, and KRAS non-G12C mutated tumours.

Variable	PFS			OS			PFS			OS		
	HR	95%CI	*p*	HR	95%CI	*p*	HR	95%CI	*p*	HR	95%CI	*p*
Age (years)	1.01	(1.00–1.01)	0.207	1.02	(1.01–1.03)	<0.001	1.01	(1.00–1.01)	0.214	1.02	(1.01–1.03)	<0.001
Sex												
Women	1 (ref)			1 (ref)			1 (ref)			1 (ref)		
Men	1.09	(0.93–1.27)	0.280	1.05	(0.88–1.24)	0.603	1.09	(0.93–1.27)	0.283	1.05	(0.89–1.24)	0.596
Smoking history												
Never	1 (ref)			1 (ref)			1 (ref)			1 (ref)		
Former/current	1.56	(1.19–2.05)	<0.001	1.71	(1.24–2.33)	0.001	1.56	(1.19–2.05)	0.001	1.70	(1.25–2.33)	0.001
ECOG PS												
0–1	1 (ref)			1 (ref)			1 (ref)			1 (ref)		
2	2.02	(1.51–2.71)	<0.001	2.90	(2.15–3.92)	<0.001	2.01	(1.50–2.70)	<0.001	2.91	(2.16–3.94)	<0.001
3–4	4.27	(2.56–7.14)	<0.001	6.51	(3.83–11.05)	<0.001	4.25	(2.54–7.11)	<0.001	6.56	(3.86–11.14)	<0.001
Stage												
I	1 (ref)			1 (ref)			1 (ref)			1 (ref)		
II	2.11	(1.58–2.81)	<0.001	2.09	(1.49–2.92)	<0.001	2.11	(1.58–2.82)	<0.001	2.08	(1.49–2.92)	<0.001
III	3.02	(2.28–3.99)	<0.001	2.67	(1.92–3.71)	<0.001	3.02	(2.28–3.99)	<0.001	2.67	(1.92–3.72)	<0.001
IV	4.61	(3.18–6.67)	<0.001	4.34	(2.91–6.47)	<0.001	4.61	(3.19–6.68)	<0.001	4.34	(2.91–6.47)	<0.001
Surgery												
No	1 (ref)			1 (ref)			1 (ref)			1 (ref)		
Yes	0.36	(0.25–0.52)	<0.001	0.25	(0.11–0.37)	<0.001	0.36	(0.25–0.52)	<0.001	0.25	(0.17–0.37)	<0.001
Curative RT +/− CT first line												
No	1 (ref)			1 (ref)			1 (ref)			1 (ref)		
Yes	0.74	(0.51–1.09)	0.127	0.47	(0.31–0.73)	<0.001	0.74	(0.51–1.09)	0.130	0.47	(0.31–0.73)	0.001
Palliative CT and/or RT first line												
No	1 (ref)			1 (ref)			1 (ref)			1 (ref)		
Yes	1.49	(1.19–1.88)	<0.001	1.16	(0.90–1.48)	0.250	1.49	(1.18–1.87)	0.001	1.16	(0.91–1.48)	0.236
History of TKI (any line)												
No	1 (ref)			1 (ref)			1 (ref)			1 (ref)		
Yes	0.78	(0.58–1.03)	0.081	0.65	(0.47–0.90)	0.009	0.78	(0.58–1.03)	0.081	0.65	(0.47–0.90)	0.009
KRAS status												
Wild type	1 (ref)			1 (ref)			-	-	-	-	-	-
Mutated	0.98	(0.83–1.15)	0.801	0.96	(0.80–1.15)	0.678	-	-	-	-	-	-
KRAS G12C status												
Wild type	-	-	-	-	-	-	1 (ref)			1 (ref)		
G12C	-	-	-	-	-	-	0.96	(0.77–1.19)	0.691	1.00	(0.79–1.28)	0.972
KRAS non-G12C	-	-	-	-	-	-	1.00	(0.82–1.21)	0.977	0.93	(0.75–1.16)	0.530

Abbreviations: PFS, progression free survival; OS, overall survival; ref reference; ECOG PS, Eastern Cooperative Oncology Group performance status; CT, chemotherapy; RT, radiotherapy; TKI, tyrosine kinase inhibitor; ICI, immune checkpoint inhibitor.

**Table 4 cancers-13-04294-t004:** Multivariable analyses of progression free survival and overall survival in patients stage I–IV with KRAS G12C, G12V, G12D, and G12A.

Variable	PFS			OS		
	HR	95%CI	*p*	HR	95%CI	*p*
Age at time of diagnosis	1.00	(0.99–1.02)	0.820	1.01	(0.99–1.03)	0.142
Sex						
Women	1 (ref)			1/ref)		
Men	1.16	(0.89–1.52)	0.283	1.01	(0.75–1.37)	0.940
Smoking history						
Never smoker	1 (ref)			1 (ref)		
Former/current smoker	0.93	(0.40–2.14)	0.858	0.77	(0.31–1.94)	0.582
ECOG PS						
0–1	1 (ref)			1 (ref)		
2	1.99	(1.16–3.40)	0.012	2.66	(1.53–4.61)	0.001
3–4	3.82	(1.64–8.88)	0.002	6.76	(2.86–16.00)	0.000
Stage						
I	1 (ref)			1 (ref)		
II	1.66	(1.01–2.74)	0.047	1.45	(0.80–2.60)	0.218
III	2.69	(1.66–4.35)	0.000	2.02	(1.13–3.60)	0.018
IV	2.49	(1.29–4.82)	0.007	2.06	(0.99–4.29)	0.054
Surgery						
No	1 (ref)			1 (ref)		
Yes	0.24	(0.13–0.46)	0.000	0.17	(0.08–0.34)	0.000
Curative RT +/− CT first line						
No	1 (ref)			1 (ref)		
Yes	0.53	(0.27–1.03)	0.062	0.26	(0.11–0.59)	0.001
Palliative CT and/or RT first line						
No	1 (ref)			1 (ref)		
Yes	1.53	(1.01–2.33)	0.046	1.26	(0.80–1.99)	0.314
KRAS mutation						
G12A	1 (ref)			1 (ref)		
G12C	0.76	(0.45–1.26)	0.281	0.97	(0.54–1.73)	0.907
G12V	0.79	(0.46–1.37)	0.399	0.83	(0.44–1.56)	0.561
G12D	0.76	(0.44–1.33)	0.333	0.87	(0.46–1.64)	0.659

Abbreviations: PFS, progression free survival; OS, overall survival; ref, reference; ECOG PS, Eastern Cooperative Oncology Group performance status; CT, chemotherapy; RT, radiotherapy.

**Table 5 cancers-13-04294-t005:** Multivariable analyses of progression free survival and overall survival in stage I–IV patients with KRAS mutations favouring PI3K/Akt (G12D), Raf (G12C, G12A, G13D, Q61L/H) and Ral (G12C).

Variable	PFS			OS			PFS			OS		
	HR	95%CI	*p*	HR	95%CI	*p*	HR	95%CI	*p*	HR	95%CI	*p*
Age (years)	1.02	(1.00–1.03)	0.070	1.02	(1.00–1.04)	0.027	1.02	(1.00–1.04)	0.085	1.01	(0.99–1.03)	0.200
Sex												
Women	1 (ref)			1 (ref)			1 (ref)			1 (ref)		
Men	1.00	(0.75–1.35)	0.981	0.88	(0.63–1.22)	0.431	0.90	(0.64–1.25)	0.529	0.69	(0.47–1.01)	0.054
Smoking history												
Never	1 (ref)			1 (ref)			1 (ref)			1 (ref)		
Former/current	1.02	(0.36–2.88)	0.975	0.93	(0.28–3.09)	0.911	1.31	(0.40–4.33)	0.653	1.34	(0.31–5.67)	0.695
ECOG PS												
0–1	1 (ref)			1 (ref)			1 (ref)			1 (ref)		
2	2.23	(1.32–3.76)	0.003	3.03	(1.79–5.14)	<0.001	2.77	(1.46–5.23)	0.002	4.89	(2.49–9.58)	<0.001
3–4	6.68	(2.89–15.5)	<0.001	9.21	(3.85–22.02)	<0.001	6.02	(2.41–14.99)	<0.001	10.95	(4.18–28.68)	<0.001
Stage												
I	1 (ref)			1 (ref)			1 (ref)			1 (ref)		
II	1.55	(0.88–2.71)	0.128	1.52	(0.79–2.94)	0.212	1.35	(0.71–2.58)	0.363	1.03	(0.47–2.26)	0.946
III	2.61	(1.49–4.58)	0.001	2.71	(1.45–5.06)	0.002	3.21	(1.77–5.85)	<0.001	2.61	(1.30–5.25)	0.007
IV	2.74	(1.36–5.52)	0.005	3.08	(1.50–6.32)	0.002	3.24	(1.46–7.19)	0.004	1.94	(0.80–4.71)	0.143
Surgery												
No	1 (ref)			1 (ref)			1 (ref)			1 (ref)		
Yes	0.234	(0.11–0.48)	<0.001	0.22	(0.1–0.48)	<0.001	0.28	(0.13–0.62)	0.002	0.16	(0.07–0.39)	<0.001
Curative RT +/− CT first line												
No	1 (ref)			1 (ref)			1 (ref)			1 (ref)		
Yes	0.617	(0.30–1.27)	0.189	0.35	(0.14–0.83)	0.017	0.57	(0.26–1.28)	0.173	0.22	(0.08–0.58)	0.002
Palliative CT and/or RT first line												
No	1 (ref)			1 (ref)			1 (ref)			1 (ref)		
Yes	1.56	(1.00–2.44)	0.052	1.27	(0.77–2.09)	0.350	1.15	(0.94–2.48)	0.089	1.32	(0.78–2.24)	0.302
PI3K vs. Raf												
Raf	1 (ref)			1 (ref)			-	-	-	-	-	-
PI3K	0.903	(0.64–1.28)	0.570	0.89	(0.60–1.30)	0.530	-	-	-	-	-	-
PI3K vs. RalA/B												
RalA/B	-	-	-	-	-	-	1 (ref)			1 (ref)		
PI3K	-	-	-	-	-	-	0.94	(0.65–1.35)	0.720	0.86	(0.58–1.29)	0.474

Abbreviations: PFS, progression free survival; OS, overall survival; ref, reference; ECOG PS, Eastern Cooperative Oncology Group performance status; CT, chemotherapy; RT, radiotherapy.

## Data Availability

The data presented in this study are available within the manuscript and the Appendix A.

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
