# Peer review of "The Prognostic Effect of KRAS Mutations in Non-Small Cell Lung Carcinoma Revisited: A Norwegian Multicentre Study"

_cancers, 2021, doi:10.3390/cancers13174294_

Round 1
Reviewer 1 Report
In this multicenter study, the authors investigated the associated between KRAS gene mutation status and clinical characteristics and survival in 1117 patients with stage I-IV NSCLC. Results of the multivariable analyses showed that there was we no significant difference in the time to disease progression or overall survival among any of the analyzed groups that were categorized according to KRAS status, G12C status, among KRAS mutation subtypes or mutation preference for downstream pathways. It was concluded that KRAS status or KRAS mutation subtype had no significant impact on PFS or OS.
I just have a couple of minor questions:
- Since 95.2 % of the patients with NSCLC included in this study had lung adenocarcinoma, shouldn't the authors just focus on adenocarcinoma in this study and leave out other NSCLC subtypes?
- It was documented that KRAS mutation status could be a good biomarker for response to EGFR-TKIs in patients with NSCLC. Did the authors observe similar results in the present study?
Author Response
In this multicenter study, the authors investigated the associated between KRAS gene mutation status and clinical characteristics and survival in 1117 patients with stage I-IV NSCLC. Results of the multivariable analyses showed that there was no significant difference in the time to disease progression or overall survival among any of the analyzed groups that were categorized according to KRAS status, G12C status, among KRAS mutation subtypes or mutation preference for downstream pathways. It was concluded that KRAS status or KRAS mutation subtype had no significant impact on PFS or OS.
I just have a couple of minor questions:
- Since 95.2 % of the patients with NSCLC included in this study had lung adenocarcinoma, shouldn't the authors just focus on adenocarcinoma in this study and leave out other NSCLC subtypes?
Response point 1: We did consider leaving out patients with tumours classified as large cell carcinoma (LCC), other non-squamous non-small cell carcinomas (NSCLC) and NSCLC-not otherwise specified (NSCLC-NOS).
However, in the seven patients diagnosed with “Other NSCLC”, the predominant diagnoses were subtypes of pleomorphic carcinoma. Since pleomorphic carcinomas often have an adenocarcinoma component and frequencies of KRAS and EGFR mutations similar to lung adenocarcinomas, we decided to include these patients. Two out of these seven patients had KRAS mutated tumours. KRAS and EGFR mutations also occur in LCC and we therefore also included patients diagnosed with LCC. Of the thirty-six patients with tumours classified as NSCLC-NOS, five patients had KRAS mutated tumours and one patient had ALK-rearranged tumour. Since all these tumours lacked immunohistochemical expression of squamous cell markers (cytokeratin 5/6 and p40), we decided to also include these patients.
- It was documented that KRAS mutation status could be a good biomarker for response to EGFR-TKIs in patients with NSCLC. Did the authors observe similar results in the present study?
Response: In our study, only 7 patients with KRAS mutated tumours had treatment with EGFR-TKIs, but not as first-line treatment. These patients were diagnosed with lung adenocarcinoma before 2004, when the first reports on the association between the presence of EGFR mutations and response to EGFR-TKI were published. Due to the small number of KRAS mutated patients treated with EGFR-TKI, we did not investigate the predictive role of KRAS mutations to EGFR-TKIs.

Reviewer 2 Report
Wahl et al presented a comprehensive analysis of the prognostic effect of KRAS mutations in 1117 non-squamous NSCLC patients in this Norwegian multicenter study. This potentially represents one of the largest retrospective studies of non-squamous NSCLC with all stages. The authors concluded that KRAS status or KRAS mutation subtypes do not have any significant prognostic influence on PFS or OS. This is interesting finding as a recent report finds that for colorectal cancer (CRC) patients with KRAS p.G12C both PFS and OS were worse than non-G12C patients (https://pubmed.ncbi.nlm.nih.gov/34250391/). It was also reported that patients with KRAS p.G12C also demonstrated higher rates of basal EGFR activation, which may be unique to CRC associated with the worse outcome. It was also highlighted that KRAS p.G12C patients may display innate resistance to chemotherapy compared with non-G12C patients. Although the CRC study was relatively larger but KRAS p.G12C is rarer in CRC than in NSCLC. In this report, the authors also included the specific pathway preference of KRAS mutations in the survival analyses. Despite being novel, simply regarding KRAS p.G12D as PI3K pathway-activating is over-simplifying. Despite biochemical evidence of higher affinity of KRAS p.G12D and p110α and the subsequent potential activation of the downstream pathway, the relation between KRAS p.G12D and PI3K activity remains unclear. There is more evidence of Q61 mutants associating with higher affinity with RAF. However, it does not extend to G12A or G13D mutants. As the authors noted, the KRAS mutant biology behind complicates the analyses and the interpretation. Perhaps using immunohistochemical data of the PI3K or MAPK pathway (e.g., p-AKT or p-ERK staining) could be more enlightening. However, it is worth acknowledging that obtaining such data retrospectively would be challenging. The authors also pointed out that co-mutations in other genes (such as TP53, STK11, and KEAP1) may also complicate the evaluation of the prognostic value of KRAS mutation status. It is now reported that STK11 mutations may be associated with worse patient outcome and may predict resistance to immune checkpoint inhibitor therapies. However, the authors acknowledges that they do not have sufficient molecular profiling data from the patients in the current study. Such molecular profiling data would be incredibly valuable to validate findings of prognostic factors from other studies. For examples, in recent clinical trials with sotorasib, patients with STK11 mutations may benefit more (showing better PFS and OS) than patients with WT STK11. Perhaps in future studies, the authors may focus on obtaining such molecular profiling data as they may have more prognostic value than KRAS mutation alone and shed light on resistance mechanisms to chemo/radiotherapies and targeted therapies, especially since KRAS mutant specific therapies (G12C and G12D specific inhibitors) and combinations with immune checkpoint inhibitors are being developed in the clinic.
The manuscript needs proofreading as some typos are found (e.g., in Fig. 1, it should be "NSCLC-NOS" not "NSCC-NOS" and its full term should be spelled out here, not in Table 1.)
Reviewer 3 Report
This is an interesting and clinical relevant manuscript that addresses the associations between KRAS mutant status and various clinicophathological features and survival. Although there are some limitations, the manuscript is suitable for publish in the journal.
Minor point:
A key omission that the authors did not have sufficient molecular data on co-occurring mutations in other genes that they may have impact on the microenvironment of KRAS-mutant non-small cell lung carcinoma. this is important for response or resistance to KRAS G12C inhibitor. My suggestion is that in the Introduction section: the authors should introduce the spectrum of major co-occurring genomic alterations in KRAS-mutant non-small cell lung carcinoma.
Reviewer 4 Report
The authors are trying to investigate survival amongst different types of KRAS mutations subtypes. They also compared their study with other conducted studies of non-Asian patients who have NSCLC.
The paper represents the results and found out that there is no significant difference in the time for the disease development and overall survival for any of the analyzed patients. The authors also briefly discussed patient characteristics according to the KRAS mutations, providing information about patients with no treatment possible to KRAS mutated tumour patients.
The research design seems appropriate. The authors have thoroughly discussed their topic and divided the paper efficiently into multiple sections, making it easy to read and understand. The authors have given a summary and abstract with appropriate background.
Study was on the association with patient characteristics to KRAS mutation and survival prognosis of such characteristics. Very detailed patient description and resulting tables. Large patient pool. Methodology was not very clear or explicitly said what they did, discussion helped in clarification.
The authors have discussed previous results related to KRAS and KRAS G12C. They have provided numerous studies with relevant references for their readers to understand the background of the NSCLC and related literature. The introduction briefly reviews RAS oncoproteins to explain KRAS gene encoding and communication with RAS protein.
The research design seems appropriate. The authors have thoroughly discussed their topic and divided the paper efficiently into multiple sections, making it easy to read and understand.
Also, I’d recommend some structural changes and review of grammar.
Following are my comments:
Why just non-Asian? What is Asian statistic?
Section: Introduction
What is 1418%?
"among subgroups of KRAS mutated patients and between KRAS mutation subtypes?" What does this mean? Please specify clearly.
- Patient data should all be defined in one spot, this was useful earlier for complete picture
Section: 2.2. Patient inclusion and tumour specimens
"STO, n=676) and the University Hospital of North Norway (UNN, n=293)" the Abstract said >1000 patients?"
In addition, 264 patients enrolled in the local lung cancer biobank at Oslo University Hospital (OUS) were evaluated for inclusion." How many included in the regional banks? Why is OUS listed separately and not consistent structure as other two.
"experienced lung pathologists in respective pathology departments of the three hospitals (SGFW, ER, MLI)."I thought the three are STO, UNN, and OUS? What are these?
Use number only for consistency 'Five-hundred and ninety-two"
